# ADAPTING CROSS-VIEW LOCALIZATION TO NEW AREAS WITHOUT GROUND TRUTH POSITIONS

## ABSTRACT

Given a ground-level query image, cross-view localization aims to estimate the location of the ground camera by matching the query to a geo-referenced aerial image that covers the local surroundings. Recent works have focused on developing powerful frameworks trained with ground truth (GT) locations of ground images within aerial images. However, the trained models always suffer a performance drop when applied to images in a new target area that differs from the training data. In most deployment scenarios, acquiring accurate GT location data for target-area images to re-train the network can be expensive and sometimes infeasible. In contrast, collecting images with coarse GT with errors of tens of meters is relatively easier. Motivated by this, our paper focuses on improving the generalization of a trained model by leveraging only the target area images without accurate GT. We propose a weakly-supervised learning approach based on knowledge self-distillation, namely, using predictions from a teacher model to supervise a student model with the same architecture. Our approach includes a mode-based pseudo GT generation for reducing uncertainty in pseudo GT and an outlier filtering to remove unreliable pseudo GT for student training. We validate our approach is generic by performing experiments on two recent state-of-the-art models with two benchmarks. The results demonstrate that our approach consistently and considerably boosts the localization performance in the target area.

## 1 INTRODUCTION

Visual localization, a fundamental task in computer vision and mobile robotics, aims to identify the location of a camera by leveraging the images it takes. Commonly, the image is compared to a pre-constructed map. However, building a 3D point cloud map (Sattler et al., 2016; Liu et al., 2017) or HD map (Ma et al., 2019) is expensive and laborious. On the other hand, aerial or satellite imagery provides global coverage and is easily accessible, making it a promising map source. In this work, we focus on the task of fine-grained cross-view localization, namely pinpointing the precise location of the ground camera within a geo-referenced aerial image patch covering local surroundings (Zhu et al., 2021; Xia et al., 2022). For applications such as autonomous driving, it provides a promising supplement to traditional positioning sensors, such as Global Navigation Satellite System (GNSS) positioning, especially in urban canyons where the GNSS positioning error can reach errors of tens of meters (Ben-Moshe et al., 2011).

Despite many works have been proposed for this task (Zhu et al., 2021; Xia et al., 2022; Shi & Li, 2022; Lentsch et al., 2023; Fervers et al., 2023; Xia et al., 2023; Shi et al., 2023), there is an important but unaddressed problem: *all methods suffer from a performance drop when directly deploying in a new target area, making them less applicable in a real-world setting.* As shown in Figure 1, there are two main scenarios in cross-view localization. **(1)** Same-area testing (Figure 1, green box): When the ground truth location of the ground images is available in the target area, a cross-view localization model can be trained on this data and then deployed for inference on new test images. **(2)** Cross-area testing (Figure 1, yellow box, left): When there is no ground truth in the target area, the standard pipeline trains the model on images from a different area where ground truth is available, and then the trained model is directly deployed in the target area. Because of the domain gap between the two areas, the predicted location becomes less reliable. Importantly, the cross-area scenario is more common in reality, as collecting accurate ground truth locations for training is expensive and sometimes infeasible. Recent works (Shi & Li, 2022; Fervers et al., 2023)

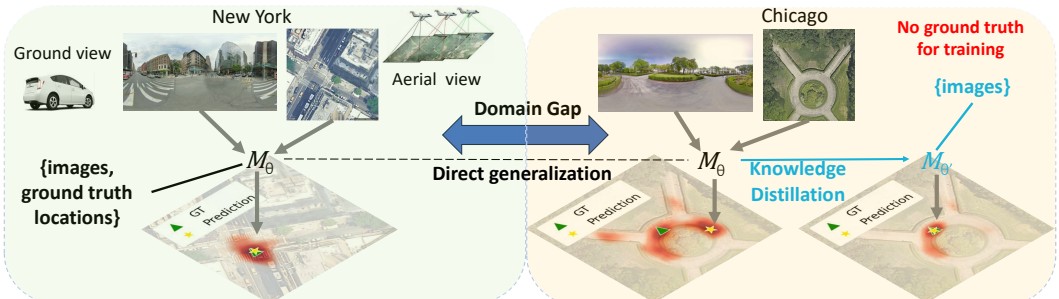

Figure 1: Standard fine-grained cross-view localization pipeline (in black), and our proposed modification (in blue). In the localization heat map at the bottom, the red color denotes the localization probability (darker: higher probability). Light green and yellow boxes denote two cities, and there is often a domain gap between images from different cities. The standard pipeline directly deploys the trained model in a different city. This often results in a performance drop, e.g. uncertain or wrong prediction. Our proposed knowledge self-distillation-based weakly-supervised learning adapts the model to the target area using only ground-aerial image pairs without ground truth positions, and it leads to better performance.

even found errors in ground truth locations in existing datasets Geiger et al. (2013); Agarwal et al. (2020); Wilson et al. (2023), demonstrating the difficulty of collecting accurate ground truth.

On the other hand, it is relatively easier to collect ground images with coarse locations, even with inaccurate GNSS measurements in urban canyons. The coarse location estimate is enough to identify the local aerial image patch that covers the ground query. Motivated by this, our goal is to *improve a pre-trained model's localization performance in the target area by leveraging only the ground-aerial image pairs in the target area, without associated ground truth locations*[1].

For our goal, we borrow insights from knowledge self-distillation (Furlanello et al., 2018; Wang & Yoon, 2021) to finetune a fine-grained cross-view localization model in a weakly supervised manner in which only coarse location is used for pairing the ground and aerial images. We use a model pre-trained from another area as the teacher model to generate pseudo ground truth for target-area images and use it to train a student model, which is initialized as a copy of the teacher model. Since the teacher's output can be uncertain in the target area, directly using it as pseudo ground truth might reinforce incorrect localization estimates and lead to sub-optimal results.

Concretely, we study an important but previously unaddressed problem in cross-view localization, *i.e.*, how to improve model performance in a new area without accurate ground truth. Our contributions are[2]: **1)** We propose a knowledge self-distillation-based weakly-supervised learning approach that considerably improves models' localization performance in a new area by only leveraging the ground-aerial image pairs without ground truth position. The proposed approach is validated on two state-of-the-art methods on two benchmarks. **2)** For methods with coarse-to-fine outputs, we investigate how to reduce the uncertainty and suppress the noise in teacher model's predictions. Using our proposed single-modal pseudo ground truth leads to a better student model than using the multi-modal heat maps from the teacher model. **3)** We propose a simple but effective method for filtering outliers in the pseudo ground truth. Training with filtered pseudo ground truth further improves the localization accuracy of the student model.

## 2   RELATED WORK

**Cross-view localization** is formulated differently depending on the use case. For large-scale coarse localization, a common formulation is image retrieval (Lin et al., 2015; Workman et al., 2015; Hu et al., 2018; Shi et al., 2019; Liu & Li, 2019; Regmi & Shah, 2019; Yang et al., 2021; Zhu et al., 2022; Shi et al., 2022a; Toker et al., 2021). In this setting, the continuous aerial imagery is divided into small patches. The ground query image's location is approximated by the retrieved patch's

---

[1]Recent models need ground camera's orientation for training. We assume the orientation is known since it can be acquired easily, *e.g.*, by the digital compass of a vehicle or a mobile phone.

[2]Our code will be released to facilitate reproducible research.

geolocation. However, for fine-grained localization, image retrieval methods need to sample the patch densely Xia et al. (2021), and it increases both computation and storage usage. Recently, there have been increasing attempts to directly estimate the precise location, sometimes together with the orientation, of the ground camera on a known aerial image patch. Zhu et al. (2021) regresses the location offset between the ground query and the aerial image from their image descriptors. Instead of regression, Xia et al. (2022) formulated the localization task as a dense classification problem to capture the multi-modal localization uncertainty. Later, this idea is extended by Xia et al. (2023) to include coarse-to-fine predictions and building orientation equivariant ground image descriptors. Several works (Shi & Li, 2022; Shi et al., 2022b; Wang et al., 2023) explored the geometry transformation between ground and aerial views. Shi & Li (2022) estimated the ground camera pose using the iterative Levenberg–Marquardt algorithm and Wang et al. (2023) made use of a deep homography estimator (Cao et al., 2022) to infer the ground camera pose. Fervers et al. (2023); Sarlin et al. (2023); Shi et al. (2023) estimated the ground camera pose by densely comparing a Bird's Eye View (BEV) representation constructed using ground images and an aerial representation. SliceMatch (Lentsch et al., 2023) took an efficient generative testing approach to select the most probable pose from a candidate set. Commonly, the final localization output is represented as a heat map (Xia et al., 2022; 2023; Fervers et al., 2023; Lentsch et al., 2023; Shi et al., 2023; Wang et al., 2023), in which the value at each location (*e.g.* pixel in the aerial image) denotes how likely the ground camera locates there, and state-of-art methods (Xia et al., 2023; Shi et al., 2023) construct the heat map in a coarse-to-fine manner.

Despite extensive methodological consideration, the performance of all the above works drops considerably when directly generalizing to images collected in an area that differs from the training set. In this work, we consider filling this gap with knowledge distillation-based weakly-supervised learning.

**Knowledge distillation** (KD), a concept initially introduced by Buciluǎ et al. (2006), aims at transmitting the knowledge acquired by a more comprehensive teacher model to a smaller student model (Wang & Yoon, 2021; Gou et al., 2021). Typically, the student model has a different architecture than the teacher model. Knowledge self-distillation, in which the teacher and student model share the same model architecture, is a special branch of KD pioneered by Born-Again Networks (Furlanello et al., 2018). The key idea is to use the student model from the previous steps to generate labels for training the model at the current step. Recent works (Zhang et al., 2019; Hou et al., 2019; Ji et al., 2021; Zhang et al., 2021; An et al., 2022) tried to use the information from deeper layers to supervise the shallower layers inside the same network. This often involves adding extra layers to generate intermediate output and using extra loss functions during training. The supervision signal at the intermediate layers can be generated from the model's final layer, from any other layers, or an ensemble of outputs from other layers (Zhang et al., 2021). KD is also used as a way for self-supervised learning for various vision tasks (Song et al., 2023; Ding et al., 2022). Despite its potential, its use for improving cross-view localization has not yet been explored.

## 3 METHODOLOGY

We first formalize the task of fine-grained cross-view localization task and our objective. After this, we introduce our proposed weakly-supervised learning approach.

### 3.1 TASK AND OBJECTIVE

Given a ground-level image $G$ and an aerial image $A$ that covers the local surroundings of $G$, the task of fine-grained cross-view localization is to determine the image coordinates $\hat{y} = (\hat{u}, \hat{v})$ of the ground camera within the aerial image $A$, where $\hat{u} \in [0, 1]$ and $\hat{v} \in [0, 1]$. Recent methods (Xia et al., 2022; 2023; Fervers et al., 2023; Lentsch et al., 2023; Shi et al., 2023) achieve this task by training a deep model $\mathcal{M}(G, A)$ which predicts a *heat map* $H$ to capture the underlying localization confidence over spatial locations, and the most certain location can be used as predicted location $y$,

$$H = \mathcal{M}(G, A), \quad y = \arg\max_{u,v}(H(u, v)). \tag{1}$$

To optimize the model's parameters $\theta_\alpha$ with respect to a model specific loss functions $\mathcal{L}_{\mathcal{M}}$, an annotated dataset of a set of $N_\alpha$ ground-aerial image pairs, $\mathbb{I}_\alpha = \{\{G_1, A_1\}, ..., \{G_{N_\alpha}, A_{N_\alpha}\}\}$, and

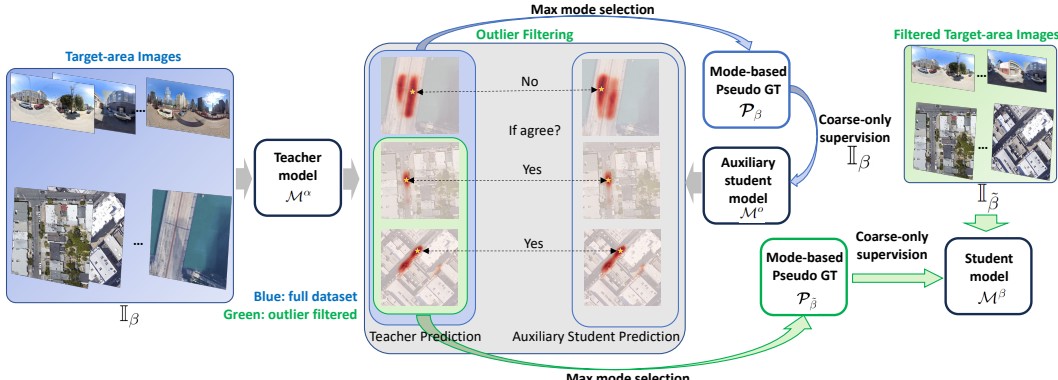

Figure 2: Overview of our proposed weakly-supervised learning approach. We first train an auxiliary student model $\mathcal{M}_o$ by knowledge distillation using the proposed mode-based pseudo-ground truth and coarse-only supervision, illustrated by blue arrows. Then, we measure the prediction differences between the teacher and the auxiliary student models for all target-area images (blue), and filter out a subset (green) for a new student model $\mathcal{M}_\beta$ supervision. The new student supervision pipeline is shown by green arrows.

their corresponding ground truth $\mathbb{Y}_\alpha = \{\hat{y}_1, ..., \hat{y}_{N_\alpha}\}$ is used,

$$\theta_\alpha = \arg\min_\theta \mathbb{E}_{(G,A)\in\mathbb{I}_\alpha, \hat{y}\in\mathbb{Y}_\alpha} \left[ \mathcal{L}_\mathcal{M}(\mathcal{M}(G, A \mid \theta), \hat{y}) \right]. \tag{2}$$

The training image set $\mathbb{I}_\alpha$ consists of samples drawn from a true distribution $\mathcal{D}_\alpha$ representing a specific geographic area $\alpha$, *i.e.* $\mathbb{I}_\alpha \overset{\text{i.i.d.}}{\sim} \mathcal{D}_\alpha$. When the model is deployed, the test image set $\mathbb{I}_{test}$ can either come from the *same area* $\alpha$, or a new environment $\beta$. As motivated before, we focus on the *cross-area* setting, namely $\mathbb{I}_{test}$ is from the target area $\beta$, *i.e.* $\mathbb{I}_{test} \overset{\text{i.i.d.}}{\sim} \mathcal{D}_\beta$. Because of the domain gap, $\mathcal{D}_\beta \neq \mathcal{D}_\alpha$, directly deploying the trained model $\mathcal{M}^\alpha := \mathcal{M}(\cdot \mid \theta^\alpha)$ on $\mathbb{I}_{test}$ as in current practice is sub-optimal. Experimental results (Xia et al., 2022; 2023) show that in the cross-area setting the predicted heat map $H$ tends to have larger uncertainty than that in the same-area setting.

Instead, we aim to improve the model performance on $\mathbb{I}_{test}$ based on two observations:

1. Collecting ground-level images with coarse location estimates in the target area is often easy, and the coarse location can be used to pair ground and aerial images. Therefore, we can have another set of images $\mathbb{I}_\beta = \{\{G_1, A_1\}, ..., \{G_{N_\beta}, A_{N_\beta}\}\}$ from the target area $\beta$, $\mathbb{I}_\beta \overset{\text{i.i.d.}}{\sim} \mathcal{D}_\beta$, without corresponding $\mathbb{Y}_\beta$. As noted before, we assume the orientation of the ground camera is known.

2. Recent state-of-the-art methods (Xia et al., 2023; Shi et al., 2023) have $K$ coarse-to-fine heat map outputs, *i.e.* $\mathbb{H} = \mathcal{M}(G, A)$ and $\mathbb{H} = \{H_1, ..., H_K\}$. The spatial resolution of the next level heat map is higher than that of the previous level, namely $\text{res}(H_{k+1}) > \text{res}(H_k)$ where $k$ is the index for the level and $\text{res}()$ returns the spatial resolution. The final predicted location then becomes $y = \arg\max_{u,v}(H_K(u, v))$. For other applications with coarse-to-fine models, encouraging shallower layers' activation to mimic deeper layers' activation can bootstrap model performance (Zhang et al., 2021).

## 3.2 PROPOSED APPROACH

Motivated by our observations, we propose a weakly-supervised learning approach based on knowledge self-distillation, which only uses a set of ground-aerial image pairs $\mathbb{I}_\beta$ from target area $\beta$ without ground truth locations. We use the trained model $\mathcal{M}^\alpha$ from the source area $\alpha$ as the teacher model to generate *pseudo ground truth* $X$ for a student model $\mathcal{M}^\beta$, which shares the same architecture as the teacher model and is initialized using the teacher model's weights $\theta_\alpha$. Here, we consider $X$ as a target heat map for the output of $\mathcal{M}^\beta$, thus with the same spatial resolution as the aerial image $A$,

Usually, the deeper layers in the model have access to more information than the shallower layers, *e.g.* the fine-grained scene layout information passed by the skipped connections, as in UNet (Ron-

neberger et al., 2015). Hence, the higher-resolution output is often more precise than the lower-resolution output. We therefore propose to follow the "Best Teacher Distillation" by Zhang et al. (2021) paradigm, and generate the pseudo ground truth $X$ from only the highest-resolution heat map predicted by the teacher model on the target domain input. A naive approach is using simply $X := H_K^{\alpha}{}^3$ from teacher output $\{H_1^{\alpha}, \cdots, H_K^{\alpha}\} = \mathcal{M}^{\alpha}(G, A)$ for any sample $\{G, A\} \in \mathbb{I}_{\beta}$.

Then, this high-resolution pseudo ground truth $X$ is down-sampled to create a set of pseudo ground truth heat maps $\mathbb{P} = \{P_1, ..., P_K\}$ to supervise the student at all levels,

$$P_k = \text{downsample}_k(X) \quad \text{s.t.} \quad \text{res}(P_k) = \text{res}(H_k). \tag{3}$$

The set $\boldsymbol{\mathcal{P}}_{\beta} = \{\mathbb{P}_1, ..., \mathbb{P}_{N_{\beta}}\}$ is the complete pseudo ground truth for image set $\mathbb{I}_{\beta}$ in the target area for training the student model, where $N_{\beta}$ is the number of the ground-aerial image pairs in $\mathbb{I}_{\beta}$.

However, it is typically observed in state-of-the-art models (Lentsch et al., 2023; Xia et al., 2023; Shi et al., 2023) that predicted heat maps in the cross-area setting contain more uncertainty than for same-area samples. The increased uncertainty results in both small positional errors, but also in more modes in the heat map yielding more outliers with high localization error. We therefore now present several strategies to reduce the teacher's uncertainty, and deal with noise and large outliers.

**Coarse-only supervision:** Standard Best Teacher Distillation suggests supervising heat maps at all levels of the student model using the pseudo ground truth. However, the spatial accuracy of $X$ is limited, and using $X$ to supervise the highest-resolution output of the student model might propagate this noise. We note the down-sampling in Equation 3 suppresses such positional noise at the lower resolution $P_k$, thus using only the lower level $P_k$ might lead to a better student model. We therefore consider to only compute the loss on $\mathbb{H}^{\beta} = \mathcal{M}^{\beta}(G, A)$ up to a certain level $K' \leq K$,

$$\mathcal{L}(\mathbb{H}^{\beta}, \mathbb{P}) = \frac{1}{K'} \sum_{k=1}^{K'} \mathcal{L}_k(H_k^{\beta}, P_k). \tag{4}$$

Here $K'$ is a hyperparameter, and $\mathcal{L}_k(H_k^{\beta}, P_k)$ a weighted sum of infoNCE losses (Oord et al., 2018) with $P_k$ being used as a weight mask,

$$\mathcal{L}_k(H_k^{\beta}, P_k) = \frac{1}{\sum P_k} \sum_{m,n} P_k^{m,n} \cdot \mathcal{L}_{\text{infoNCE}}(H_k^{\beta} \mid (m,n)). \tag{5}$$

In Equation 5, $\mathcal{L}_{\text{infoNCE}}(H_k^{\beta} \mid (m,n))$ denotes the infoNCE loss interpreting $H_k^{\beta}$ as a classification output, location $(m,n)$ as the positive class, and all other locations as the negative class.

**Mode-based Pseudo Ground Truth:** Rather than using $H_K^{\alpha}$ directly as pseudo ground truth $X$, we propose to create a 'clean' pseudo ground truth $X$ that only represents its mode $y^{\alpha} = \arg\max(H_K^{\alpha})$. We thus provide the student with a training objective that represents less uncertainty for the target domain input than its teacher. Still, it is common when training fine-grained cross-view localization models, to apply Gaussian label smoothing (Xia et al., 2022; Fervers et al., 2023) even with reliable ground truth to aid the learning objective and increase robustness to remaining errors in the annotation (Müller et al., 2019). We similarly apply some Gaussian label smoothing centered at $y^{\alpha}$,

$$X(u,v) = \mathcal{N}((u,v) \mid y^{\alpha}, I_2\sigma^2), \quad \text{res}(X) = \text{res}(A). \tag{6}$$

In Equation 6, standard deviation $\sigma$ is a hyperparameter and $I_2$ is a 2D identity matrix.

**Outlier Filtering:** Recent deep learning advances (Oquab et al., 2023) highlighted the importance of using curated data. Motivated by this principle, we prefer having few but reliable samples of the target domain, over having more samples but with potentially large errors in the ground truth. The *Mode-based Pseudo Ground Truth* could force a sample's ground truth to commit to a wrong (outlier) location, therefore we seek to filter out such samples.

We here make another observation: samples where the predicted locations $y^{\alpha}$ of a teacher and $y^{\beta}$ student greatly differ, the teacher's prediction was more likely to be outlier compared to the true location $\hat{y}$, as we will demonstrate in our experiments. Therefore, we propose to first train another auxiliary student model on all data from the target domain, and compare its prediction to the

---

[3]Note that we use superscript $\alpha$ to indicate output generated by model $\mathcal{M}^{\alpha}$.

teacher's to identify stable predictions with little change in the predicted location. Then, we only use those reliable non-outlier samples to train the final student model.

Concretely, we first optimize the auxiliary student model $\mathcal{M}^o$ on all $\mathbb{I}_\beta$ with $\mathcal{P}_\beta$ using,

$$\theta_o = \arg\min_\theta \mathbb{E}_{\{G,A\}\in\mathbb{I}_\beta, \mathbb{P}\in\mathcal{P}_\beta} \left[ \mathcal{L}(\mathcal{M}(G, A \mid \theta), \mathbb{P}) \right]. \tag{7}$$

Then, we calculate the L2-distance $d^{\alpha,o} = \|\mathbf{y}^\alpha - \mathbf{y}^\mathbf{o}\|_2$ between the image coordinates predicted by $\mathcal{M}^\alpha$ and $\mathcal{M}^o$ to find the potential unreliable $\mathbb{P}$. The resulting distance set $\mathbb{D} = \{d_1^{\alpha,o}, ..., d_{N_\beta}^{\alpha,o}\}$ is used to find top-$T\%$ samples in $\mathbb{I}_\beta$ that have the smallest $T\%$ distance $d^{\alpha,o}$. Denoting the resulting image set as $\mathbb{I}_{\tilde{\beta}}$ and corresponding pseudo ground truth as $\mathcal{P}_{\tilde{\beta}}$, the final student model $\mathcal{M}^\beta$ is optimized using Equation 7 after substituting $\mathbb{I}_\beta$ with $\mathbb{I}_{\tilde{\beta}}$ and $\mathcal{P}_\beta$ with $\mathcal{P}_{\tilde{\beta}}$.

## 4 EXPERIMENTS

We first introduce the two used datasets, followed by our evaluation metrics. Then we discuss two state-of-the-art methods (Xia et al., 2023; Shi et al., 2023) used for testing our proposed approach and our implementation details. After this, we provide the test results and a detailed ablation study.

### 4.1 DATASETS

We use two common cross-view localization datasets, VIGOR (Zhu et al., 2021) and KITTI (Geiger et al., 2013), and focus on their cross-area split.

**VIGOR** dataset contains ground-level panoramic images and their corresponding aerial images collected in four US cities. In its cross-area split, the training set contains images from two cities, and the test set is collected from the other two cities. We use the training set to train the teacher model and focus on the cross-area setting in our experiments. To compare direct generalization and our proposed weakly-supervised learning, we conduct a 70%, 10%, and 20% split on the original cross-area test set to create our weakly-supervised training set (no ground truth), validation set, and test set. Weakly-supervised training set does not have ground truth labels and is used for training the student model. We use the validation set to find the stopping epoch for training, as well as the ablation study. Our test set is used for benchmarking our method. We use the improved VIGOR labels provided by Lentsch et al. (2023).

**KITTI** dataset contains ground-level images with a limited field-of-view, collected in Germany. We use the aerial images provided by Shi & Li (2022) and adopt their cross-area setting, namely, the training and test images are from different areas. Similar to our settings on the VIGOR dataset, we use the training set to train the teacher model and then split the original cross-area test set into 70%, 10%, and 20% for weakly-supervised training of the student model, validation, and testing.

### 4.2 EVALUATION METRICS

We measure the displacement error in meters, $\epsilon$, between the predicted location and the ground truth location, *i.e.*, $\epsilon = s\|y - \hat{y}\|_2$, where $s$ is the scaling from image coordinates to real-world Euclidean coordinates. Then, we use mean and median displacement error as our evaluation metrics. Since ground-level images in the KITTI dataset have a limited field-of-view, we further decompose the displacement errors into errors in the longitudinal direction (along the camera's viewing direction, typically along the road), and errors in the lateral direction (perpendicular to the viewing direction).

### 4.3 USED STATE-OF-THE-ART METHODS

Two state-of-the-art methods, Convolutional Cross-View Pose Estimation (CCVPE) (Xia et al., 2023) and Geometry-Guided Cross-View Transformer (GGCVT) (Shi et al., 2023) are used to test our proposed weakly-supervised learning approach. Both methods were proposed for fine-grained cross-view localization and orientation estimation, and have a coarse-to-fine architecture. CCVPE has two separate branches for localization and orientation prediction. GGCVT uses an orientation estimation block before its location estimator. In this work, we use them for localization only. CCVPE has seven levels of heat map outputs, in which the first six heat maps are 3D, with the first

two dimensions for localization and the third dimension for orientation. The last heat map is 2D. GGCVT has three levels of 2D heat map outputs.

## 4.4 IMPLEMENTATION DETAILS AND OUR SETTINGS

We use the code released by the authors of CCVPE (Xia et al., 2023) and GGCVT (Shi et al., 2023) for model implementation. Our introduced loss is used for weakly-supervised learning of the student model. For CCVPE's 3D heat map output, we simply lift the pseudo ground truth heat map $P_k$ to 3D using the known orientation as done by Xia et al. (2023). Following the two model's default settings, we use a batch size of 8 for CCVPE and 4 for GGCVT, and use the learning rate of $1 \times 10^{-4}$ with Adam optimizer (Kingma & Ba, 2014) for both models.

For weakly-supervised learning, we tune our hyperparameters, $K'$, $T$, and $\sigma$ on the VIGOR validation set. For CCVPE, we find that including the first two levels of losses, $i.e.$ $K' = 2$, and $T\% = 80\%$ samples gives the lowest mean localization error. For GGCVT, we use all three levels of losses, $i.e.$ $K' = 3$, and $T\% = 70\%$. We use $\sigma = 4$ (pixels) for both methods. On the KITTI dataset, we directly use the same setting.

## 4.5 TEST RESULTS

We compare the learned student models to teacher models (baselines) on the cross-area test set of VIGOR and KITTI datasets. Previous state-of-the-art was set by directly deploying CCVPE and GGCVT teacher models to the target area. On the VIGOR dataset, Table 1 top, the performance of student models trained using proposed weakly-supervised learning surpasses baselines by a large margin. For the CCVPE model, it reduces the mean and median error by 20% and 15% when the orientation of test ground images is unknown. GGCVT needs orientation-aligned ground panorama-aerial pairs for inference. In this case, our weakly-supervised learning reduces 16% and 5% mean and median error for GGCVT. Without extra hyperparameter tuning, we directly use our proposed weakly-supervised learning approach to train models on the KITTI dataset, and it again improves the overall localization performance for both models, see Table 1 bottom.

Table 1: Evaluation on VIGOR and KITTI test set. **Best in bold.** Baseline models are teacher models (previous state-of-the-art). "Student" denotes using our proposed weakly-supervised learning without ground truth labels. On VIGOR, we provide test results for both known and unknown orientation cases. GGCVT only works with panoramas with known orientation. On KITTI, we test with known orientation.

| VIGOR, cross-area test | Known orientation | | Unknown orientation | |
|---|---|---|---|---|
| | mean (m) | median (m) | mean (m) | median (m) |
| CCVPE (Xia et al., 2023) | 4.38 | 1.76 | 5.35 | 1.97 |
| CCVPE student (ours) | **3.85** ($\downarrow 12\%$) | **1.57** ($\downarrow 11\%$) | **4.27** ($\downarrow 20\%$) | **1.67** ($\downarrow 15\%$) |
| GGCVT (Shi et al., 2023) | 5.19 | 1.39 | - | - |
| GGCVT student (ours) | **4.34** ($\downarrow 16\%$) | **1.32** ($\downarrow 5\%$) | - | - |
| KITTI, cross-area test | Longitudinal error | | Lateral error | |
| | mean (m) | median (m) | mean (m) | median (m) |
| CCVPE (Xia et al., 2023) | 6.55 | 2.55 | 1.82 | **0.98** |
| CCVPE student (ours) | **6.18** ($\downarrow 6\%$) | **2.35** ($\downarrow 8\%$) | **1.76** ($\downarrow 3\%$) | **0.98** ($\downarrow 0\%$) |
| GGCVT (Shi et al., 2023) | 9.27 | 4.66 | 2.19 | 0.85 |
| GGCVT student (ours) | **8.56** ($\downarrow 8\%$) | **4.35** ($\downarrow 7\%$) | **1.90** ($\downarrow 13\%$) | **0.79** ($\downarrow 7\%$) |

Next, we visualize two samples where the student model improves over the teacher model. A typical case is shown in Figure 3 first row, in which the teacher model has a multi-modal prediction, and the peak is located in a wrong mode. The student model learned to better weigh the modes after adapting to the target environment. As shown in the second example in Figure 3, sometimes, even though the teacher model's heat map does not capture the correct location, the student model can still identify it. In this case, the student model might learned discriminative features from other samples in this area to localize the ground camera. This highlights the effectiveness of the knowledge-distillation process for cross-area evaluation.

In what follows, we analyze the behavior of our teacher and student models using the CCVPE method. The same analysis using the GGCVT method is presented in the Appendix. Figure 4a

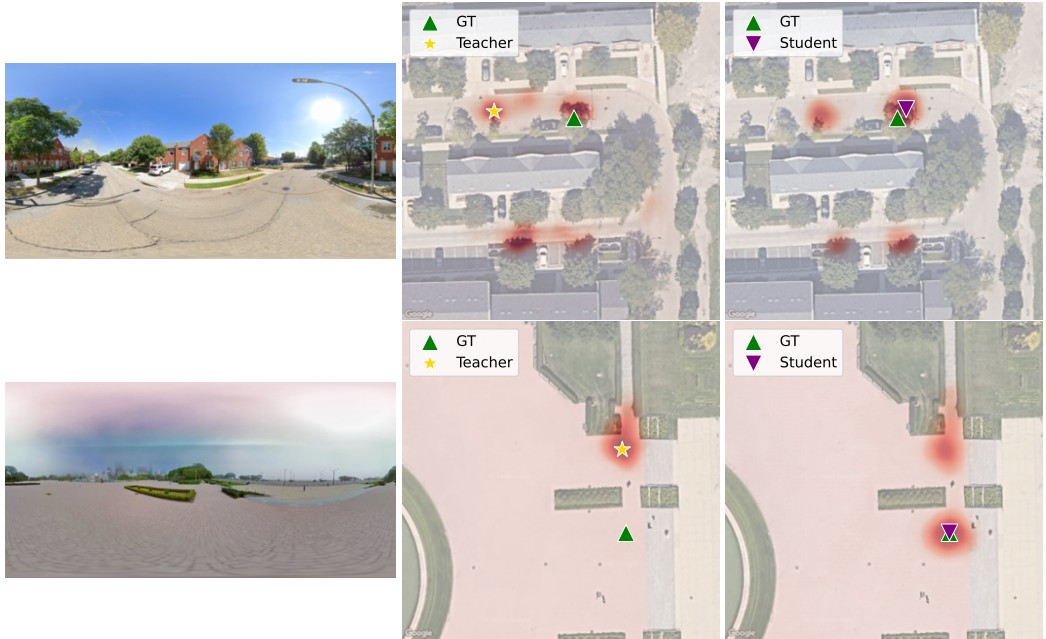

Figure 3: CCVPE teacher and student model's predictions on VIGOR test set. The red color denotes the localization probability (a darker color means a higher probability).

compares the teacher model and the auxiliary student model on the training set $\mathbb{I}_\beta$. For both figures, the horizontal axis denotes the teacher and student prediction differences. The vertical axis indicates the prediction errors by the teacher model (left) and the auxiliary student model (right). From the left figure, there is a large portion of data whose teacher prediction error equals the teacher-and-student prediction differences, indicated by the scattered diagonal line. After training, this line is less prominent in the student model's prediction visualization (right), where many samples from the diagonal line move to the bottom part of the figure with a smaller prediction error (*e.g.*, 0–5m). This indicates that the auxiliary student model has corrected various wrong predictions from the teacher model to the correct location and confirms our designed outlier filtering can identify the outliers in the teacher's prediction. The comparison between the teacher model and the final student model trained after outlier detection evaluated on the target test set $\mathbb{I}_{test}$ is visualized in Figure 4b, where the same phenomenon is observed. The diagonal line in the teacher model's predictions is less prominent in the student model's predictions, demonstrating the student model successfully reduces the localization error for many samples compared to the teacher model.

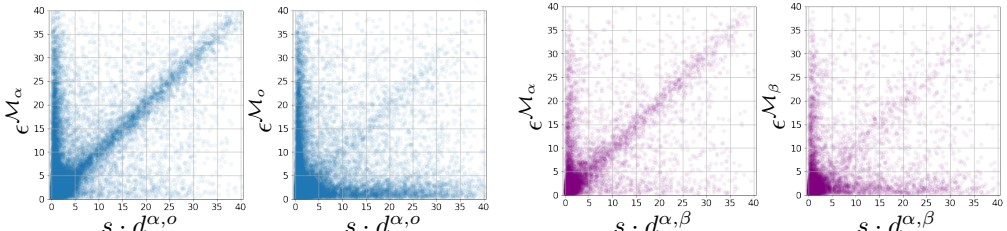

(a) Teacher *vs.* Auxiliary student models on $\mathbb{I}_\beta$      (b) Teacher *vs.* Final student models on $\mathbb{I}_{test}$

Figure 4: CCVPE model, relation between error $\epsilon$ and change $d$ in predicted locations from teacher and student models on VIGOR.

## 4.6 ABLATION STUDY

We conduct an extensive ablation study to validate the effectiveness of our proposed designs. We denote the following: (a) **Teacher** (baseline): directly deploy the teacher model $\mathcal{M}^\alpha$ in the target area. (b) **Student-M-OF**: student model trained using teacher's heat maps, no mode-based pseudo

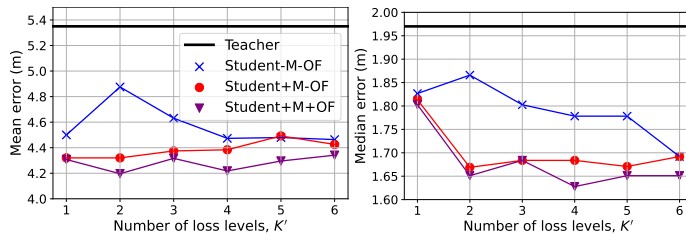 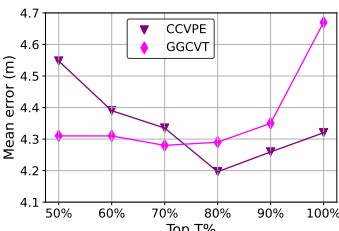

Figure 5: Ablation study results on the proposed mode-based pseudo ground truth, outlier filtering, and different coarse levels that needs supervision in the teacher-student localization distillation, with the baseline as CCVPE.

Figure 6: Comparison with different $T$ in the proposed outlier filtering. $T = 100\%$ means no outlier filtering is used.

Table 2: Ablation study on proposed components on validation set, GGCVT model. **Best in bold.**

| Error (m) | Teacher | Student-M-OF | Student+M-OF | Student+M+OF |
|---|---|---|---|---|
| mean | 5.16 | 5.34 | 4.67 | **4.28** |
| median | 1.40 | 1.48 | 1.32 | **1.28** |

ground truth, no outlier filtering. (c) **Student+M-OF**: student model trained using mode-based pseudo ground truth, no outlier filtering. (d) **Student+M+OF** (proposed): student model trained using mode-based pseudo ground truth with outlier filtering, *i.e.* $\mathcal{M}^\beta$.

The performance of these ablation variants when supervising different levels of student predictions of the CCVPE is shown in Figure 5. It can be seen that the proposed mode-based pseudo ground truth (+OM) and outlier filtering (+OF) both improve the performance, and the final version, *Student+M+OF*, achieves the best results, no matter how many prediction levels of the student model is supervised. Supervising the first $K' = 2$ levels of a CCVPE student model gives the lowest mean error on the validation set. We also tuned $K'$ for GGCVT and found that supervising all three levels, *i.e.* $K' = 3$ gives the best results. The effectiveness of the proposed mode-based pseudo ground truth (+OM) and outlier filtering (+OF) on GGCVT is verified in Table 2.

Figure 6 shows the ablation study results on different percentage values $T$ in our outlier detection. Best CCVPE and GGCVT student models appear at $T = 80\%$ and $T = 70\%$. In general, there is a trade-off between the quality and quantity of data. When too little data is kept, there is a risk of model overfitting. Filtering out some detected outliers ($20\% \sim 30\%$) improves the quality of the data and can result in better model performance. This suggests that, in practice, blindly increasing the data amount without checking its quality might negatively influence the model performance.

## 5 CONCLUSION

This paper focuses on improving the localization performance of a trained cross-view localization model in a new target area without accurate ground truth locations. For this, we have proposed a knowledge self-distillation-based weakly-supervised learning approach that only uses a set of ground-aerial image pairs from the target area without ground truth locations. Extensive experiments were conducted to study how to generate pseudo ground truth for student model learning. We found that it is better to select the predominant mode in the teacher model's predictions than directly using its heat maps. Furthermore, supervising coarse-level predictions of a student model using the down-sampled teacher model's high-resolution predictions can suppress the positional noise and might lead to a small boost in the student model's performance. The specific number of coarse levels depends on different models. Last but not least, we demonstrate unreliable target domain samples can be filtered out by comparing teacher and student models, which motivates using an auxiliary student model to curate the data. Training a final student model on the filtered data further improves the localization accuracy. Our proposed approach has been validated on two state-of-the-art methods on two benchmarks. It achieves a consistent and considerable performance boost over the previous standard that directly deploys the trained model in the new target area. We advocate researchers in this community to pay attention to this critical problem and explore new innovative solutions.

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

## A    SAMPLE DISTRIBUTION IN TERMS OF ERROR DIFFERENCE BETWEEN TEACHER AND STUDENT MODEL PREDICTIONS

We visualize the histogram distribution of samples in the target area test set in terms of the difference between the teacher and the final student model prediction errors, see Figure 7. The number of samples where the student model has a smaller error than the teacher model is larger than those where the teacher model outperforms the student model. This demonstrates the student model reduces the error for the majority of samples.

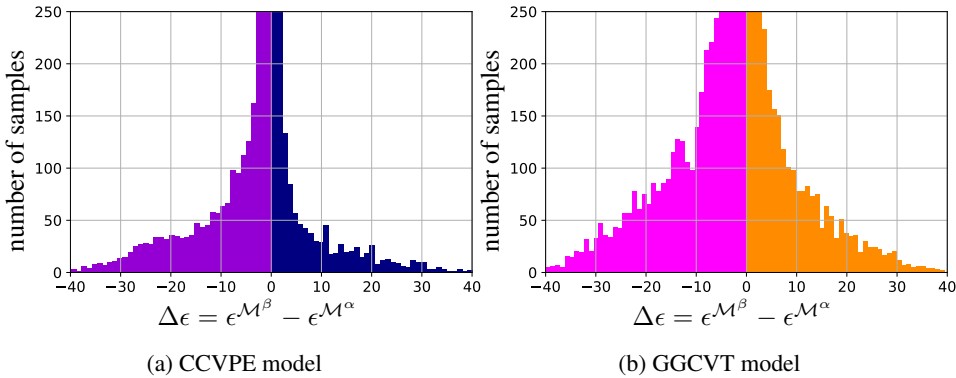

(a) CCVPE model

(b) GGCVT model

Figure 7: Change in error between teacher $\mathcal{M}^\alpha$ and student model $\mathcal{M}^\beta$ on VIGOR test set $\mathbb{I}_{test}$. Purple and Magenta region: The student model has smaller errors $\epsilon$. Navy and Orange region: The teacher has smaller errors.

## B SUPPLEMENTED BEHAVIOR ANALYSIS OF THE TEACHER AND STUDENT MODELS ON THE BASELINE OF GGCVT

We supply the error change analysis of the teacher and student models on the baseline of GGCVT in Figure 8. The figure illustrates similar observations with the baseline of CCVPE. Specifically, there is a large portion of data where the teacher model's prediction has the same error as the prediction differences between the teacher and student models' predictions, as indicated by the scattered diagonal line in the left figure of Figure 8a and Figure 8b. However, this line is less prominent in the student models' predictions, as illustrated in the right figure in Figure 8a and Figure 8b. This demonstrates that the student model supervised by our proposed strategy successfully reduces the teacher model's prediction errors in many cases.

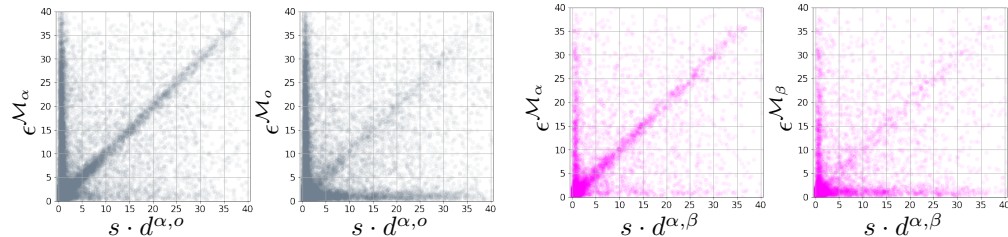

(a) Teacher *vs*. Auxiliary student models on $\mathbb{I}_\beta$      (b) Teacher *vs*. Final student models on $\mathbb{I}_{test}$

Figure 8: GGCVT model, relation between error $\epsilon$ and change $d$ in predicted locations from teacher and student models on VIGOR.

## C EXTRA QUALITATIVE RESULTS OF TEACHER AND STUDENT MODELS

More visualization of teacher and student models' predictions is shown in Figure 9.

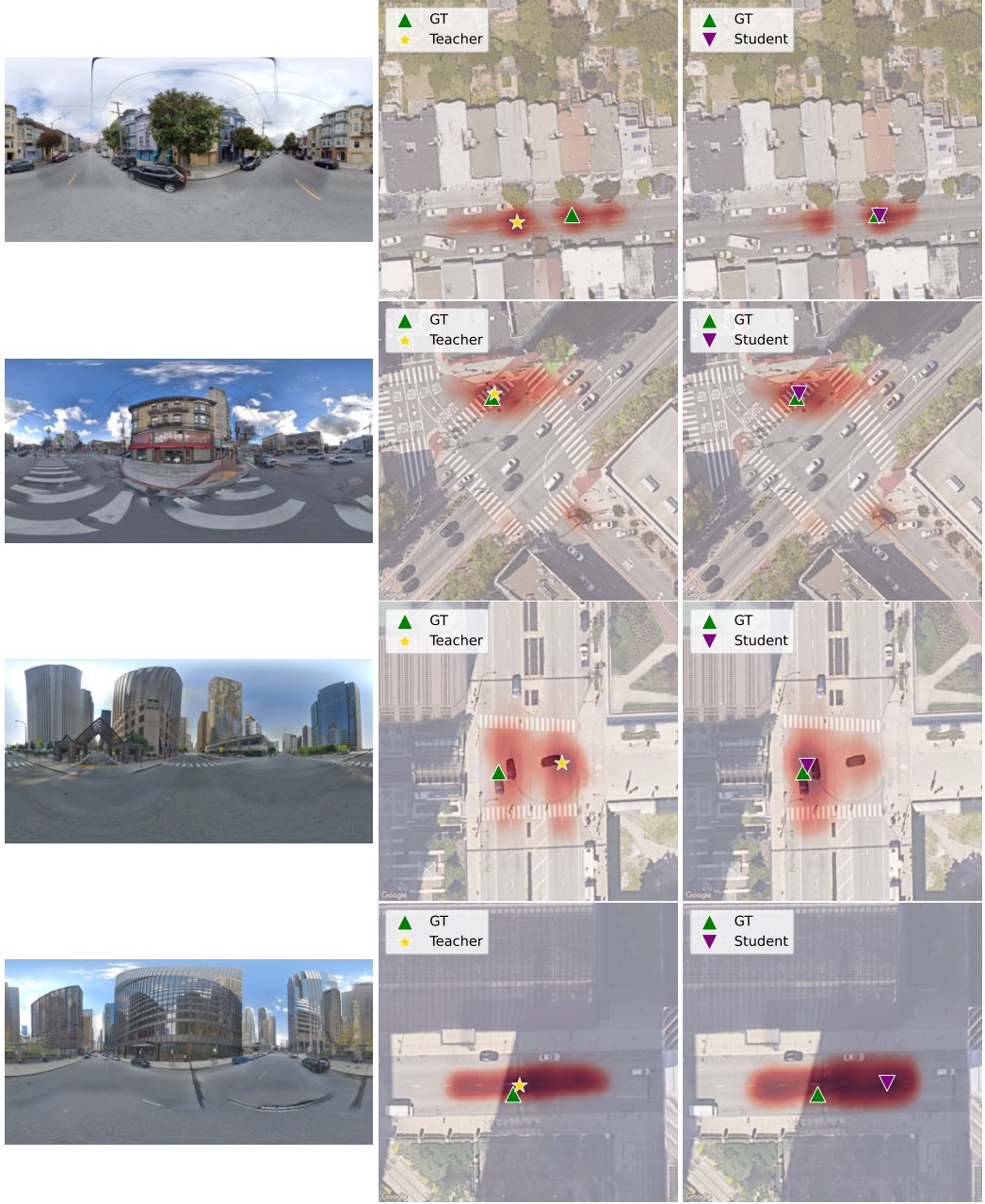

Figure 9: Teacher and student model's predictions on VIGOR test set. The red color denotes the localization probability (a darker color means a higher probability). First three: success cases. Last: failure case.

