# OpenReview forum: "Adapting Cross-View Localization to New Areas without Ground Truth Positions"
_ICLR.cc/2024/Conference — ICLR 2024 Conference Withdrawn Submission_

### Official Review · Reviewer_8CJA · 2023-10-30

**Soundness:** 2 fair
**Presentation:** 3 good
**Contribution:** 2 fair
**Rating:** 5
**Confidence:** 4

**Summary:**

This paper deals with the cross-view localization problem and considers the domain transfer issue for new areas. It proposes a teacher-student pipeline to improve the generation ability of existing works. This work assumes that the images of the target area are available, but there is no label. The experiments are conducted on VIGOR and KITTI dataset.

**Strengths:**

+ Generalization is very important for the localization system and the work is well-motivated.
+ The proposed method is evaluated on two widely used datasets with detailed evaluation metrics.
+ The writing is easy to follow.
+ The qualitative results look good. The ablation study and analysis are also well presented.

**Weaknesses:**

-	The proposed method is more like semi-supervised learning rather than weakly-supervised learning, as it generates pseudo labels for training on the target area. Both knowledge distillation and domain adaptation have studied similar problems.
-	The experimental setting could be improved to better support the motivation, i.e. generalization on new areas. The current setting seems to split the images of the target area into several parts. Although the label is not provided, the ground-truth pairs still exist in the training set of the target area, which is not the case for real-world applications. It is very common that some query images may not be covered by the reference images in the target areas. In other words, some images may not have the correct match in the training set of the target area.

-	The performance improvement is limited, especially on KITTI dataset.

-	The computation cost of the proposed method is not discussed. Given that previous methods have achieved high accuracy on these datasets and the proposed method introduces additional computational cost, It is important to discuss the trade-off between the performance improvement and the additional computational cost.

**Questions:**

See the weaknesses.

---

> ### Author Response · Authors · 2023-11-13
>
> We appreciate your feedback. Hereby, we address some concerns.
>
> **Reviewer**: “*The experimental setting could be improved to better support the motivation, i.e. generalization on new areas. The current setting seems to split the images of the target area into several parts. Although the label is not provided, the ground-truth pairs still exist in the training set of the target area, which is not the case for real-world applications. It is very common that some query images may not be covered by the reference images in the target areas. In other words, some images may not have the correct match in the training set of the target area*”:
>
> **Author response**: We agree that experimental settings should support the motivation, i.e. generalization to new areas. Because of this, we do not use ground truth position for the training in the target area. As motivated in the task setting (please refer to our reply to Reviewer Tcvm), a coarse GNSS prior is often available, also for the inference time in real world settings, such as autonomous driving and outdoor robotics. Hence, the task of fine-grained cross-view localization do assume the ground-aerial image pairs are known.
>
>
>
> **Reviewer**: “*The proposed method is more like semi-supervised learning rather than weakly-supervised learning, as it generates pseudo labels for training on the target area. Both knowledge distillation and domain adaptation have studied similar problems*”:
>
> **Author response**: We use the term “weakly-supervised learning” since the task of fine-grained cross-view localization does provide ground-aerial image pairs. There exists a rough localization prior of the ground image. The term “weakly” indicates the existence of the rough localization prior.
>
> The term “semi-supervised” would instead imply we have localization labels for some locations, and no labels for others. We see generating pseudo labels for training as a form of *self-supervised learning*. However, since self-supervised learning does not imply we know the ground-aerial image pairs, we find “weakly-supervised learning” is more appropriate.

---

### Official Review · Reviewer_Tcvm · 2023-10-30

**Soundness:** 2 fair
**Presentation:** 2 fair
**Contribution:** 1 poor
**Rating:** 3
**Confidence:** 4

**Summary:**

In this paper, the authors propose a weakly-supervised learning approach using knowledge self-distillation to improve the cross-view localization performance in new target areas without accurate ground truth positions. However, the paper is flawed in terms of writing, innovations, and experiments.

**Strengths:**

+ This article presents a self-distillation framework to enhance the performance of models across domains.

**Weaknesses:**

- There is a lack of mathematical analysis as to why self-distillation frameworks are able to improve the fine-grained localization by only using coarse labels from target domain. Intuitive explanations and visualisation diagrams alone are not convincing enough.
- The ablation experiments are insufficient. Lacking of comparison with domain adaption methods, and enhancements over baseline methods do not entirely come from pseudo label supervision by the teacher.
- The test results are insufficient. Lack of indicators of the success rate of matching between ground and aerial images e.g. R@1, R@5, Hit Rate. A direct comparison of metre-level localization accuracy in the absence of a matching success rate is meaningless.

**Questions:**

1.	The proposed introduces the coarse-grained labels from target domain so it is considered a domain adaptation approach. Is the boost due to having seen the distribution of target domains, or is it due to the weak supervision of the pseudo-labels? The addition of ablation experiments to compare other domain adaptation methods is recommended.
2.	Why the poor model (the teacher) can lead good model (the student) in a good direction? Hopefully the authors will give solid mathematical derivations rather than intuitive descriptions and visualisations. This is because visualised heatmaps may simply come from success cases, which cannot be controlled at the time of review.
3.	For the self-distillation approach, it is necessary to maintain the teacher model and the student model in the memory, which is not too demanding in terms of computational resources? I am concerned about the ease of reproducing the method proposed in this paper and suggest that the computational cost be given.
4.	Lacking of indicators of the success rate of matching between ground and aerial images e.g. R@1, R@5, Hit Rate. A direct comparison of metre-level localization accuracy in the absence of a matching success rate is meaningless.
5.	As the most important general framework diagram, the font size in Figure 2 is too small and the overall flow is not clear and concise. It is recommended that it be redrawn.

**Details Of Ethics Concerns:**

I do not have the Ethics Concerns.

---

> ### Author Response · Authors · 2023-11-13
>
> Thank you for your valuable feedback.
>
> First of all, we would like to clarify our task setting and then address some related concerns.
>
> **Task setting**:
>
> Our paper focuses on fine-grained cross-view localization [Xia et al., 2022; Shi & Li, 2022; Lentsch et al., 2023; Fervers et al., 2023; Xia et al., 2023; Shi et al., 2023], a newly emerging task distinct from cross-view image retrieval-based localization. This task assumes a rough localization prior (e.g., noisy GNSS positioning or temporal filtering) is already available, which is realistic for real-world applications like autonomous driving. The localization prior can identify an aerial image that covers local surroundings. Therefore, the task in our submission and these previous works aims to precisely locate the ground camera within a known aerial image, and *does not*  concern retrieving corresponding aerial images.
>
>
> **Reply to the concerns**: Weakness “*The test results are insufficient. Lack of indicators of the success rate of matching between ground and aerial images e.g. R@1, R@5, Hit Rate. A direct comparison of metre-level localization accuracy in the absence of a matching success rate is meaningless.*” and the question “*Lacking of indicators of the success rate of matching between ground and aerial images e.g. R@1, R@5, Hit Rate. A direct comparison of metre-level localization accuracy in the absence of a matching success rate is meaningless*”:
>
> As we clarified in our task setting, since the ground-aerial image pairs are known, recall *cannot be used* as an evaluation metric. Instead, localization error in meters is the key metric that measures how close the estimated location is to the ground truth location. Note that, prior work in this domain [Xia et al., 2022; Shi & Li, 2022; Lentsch et al., 2023; Fervers et al., 2023; Xia et al., 2023; Shi et al., 2023] also measures localization error in meters instead of ground-aerial image retrieval hit rate.

---

### Author Response · Authors · 2023-11-17

We would like to thank the reviewers again for their time and feedback.
We decided to retract the paper and further improve it based on the reviews.